

# Beyond top-k: knowledge reasoning for multi-answer temporal questions based on revalidation framework

Junping Yao[*], Cong Yuan[*], Xiaojun Li, Yijing Wang and Yi Su

Xi'an Research Inst. of High-Tech, Xi'an, Shaanxi, China
[*] These authors contributed equally to this work.

## ABSTRACT

Answer sorting and filtering are two closely related steps for determining the answer to a question. Answer sorting is designed to produce an ordered list of scores based on Top-k and contextual criteria. Answer filtering optimizes the selection according to other criteria, such as the range of time constraints the user expects. However, the unclear number of answers and time constraints, as well as the high score of false positive results, indicate that the traditional sorting and selection methods cannot guarantee the quality of answers to multi-answer questions. Therefore, this study proposes MATQA, a component based on multi-answer temporal question reasoning, using a re-validation framework to convert the Top-k answer list output by the QA system into a clear number of answer combinations, and a new multi-answer based evaluation index is proposed for this output form. First, the highly correlated subgraph is selected by calculating the scores of the boot node and the related fact node. Second, the subgraph attention inference module is introduced to determine the initial answer with the highest probability. Finally, the alternative answers are clustered at the semantic level and the time constraint level. Meanwhile, the candidate answers with similar types and high scores but do not satisfy the semantic constraints or the time constraints are eliminated to ensure the number and accuracy of final answers. Experiments on the multi-answer TimeQuestions dataset demonstrate the effectiveness of the answer combinations output by MATQA.

## INTRODUCTION

A high-quality question answering (QA) model (*Jia et al., 2018*) is sensitive to constraints on semantic quantitative boundaries of input questions. Mainstream question answering approaches intentionally reduce the task to a "one best answer per question" scheme. But in practice, many temporal problems are open-ended and ambiguous, with multiple valid answers (or groups of answers), and often all of these answers must be captured so as to answer one question (*Rubin et al., 2022*). *Min et al. (2020)* pointed out that over 50% of the query intent in Google search is ambiguous. In order to show strong reasoning ability, the question answering model not only needs to give the answer with high confidence but

Corresponding author
Xiaojun Li, xi_anlxj@126.com

also the exact number of answers. Nevertheless, the existing question answering systems can only obtain the Top-k list of a single answer by scoring ranking (*Wang et al., 2021*). When there are multiple valid answers to a temporal question, users cannot directly obtain valid solutions with high accuracy and accurate numbers.

Multi-answer reasoning stems from reading comprehension. Currently, multi-answer reasoning is based on unstructured text databases and aims to retrieve all answers from multiple passages that satisfy the intention of a question. Limited by the ambiguity of natural language, questions can be interpreted with multiple meanings, so multiple answers will be recalled from the text. Limitations of existing work (*Rubin et al., 2022*; *Min et al., 2020*; *Shao & Huang, 2022*) concern various forms of paragraph parsing and question and ambiguous answer matching. Retrieving and reading paradigm is the major method of text paragraph multi-answer reasoning. It involves the correct reasoning of long sequences of paragraphs in the computation process, with restrictions on both the maximum number of paragraphs supported by hardware and their mutual interaction. For example, AMBIGNQ (*Min et al., 2020*) utilizes the BERT dual encoding model for retrieving and reordering 100 paragraphs. It concatenates the question with the top paragraph to generate the answer in an end-to-end system. *Shao & Huang (2022)* used the "recall-revalidation" framework to avoid the problem of multiple answers sharing a limited reading budget by separating the reasoning process of each answer and to better verify the answer with re-found evidence. *Liu et al. (2021)* alleviated the error propagation problem by explicitly modeling three matching granularities of paragraph recognition, sentence selection and answer extraction through MGRC, an end-to-end reading comprehension model.

Multi-answer reasoning based on knowledge base is in its infancy. *Moon et al. (2022)* in 2022 proposed RxWhyQA, a clinical question answering dataset for multi-answer questions, and pointed out that clinical reasoning and decision making are still constrained by multi-answer questions. In the same year, *Zhong et al. (2022)* proposed RoMQA, a benchmark for multi-evidence, multi-answer question answering. Despite revealing the shortcomings of existing zero-sample, small-sample learning and supervised learning schemes on this benchmark, they failed to propose a clear solution. In the field of temporal question answering, there is no perfect method to solve the multi-answer reasoning problem. This study aims to extend the multi-answer question answering to the field of temporal knowledge question answering. Based on the knowledge base, the main work is to ensure the numerical quality of valid answers to temporal questions. Although the existing unstructured question answering (*Cao et al., 2021*) and knowledge-based question answering schemes have achieved good results, there are still the following new challenges in the field of multi-answer temporal question reasoning:

**The number of answers is undetermined.** In practice, there exists a class of multi-answer problems in which the answer consists of multiple entities or attributes. For example, in temporal question answering, there are usually more than one candidate answer to be accepted within a given time interval. However, the traditional Top-k list only shows the ranking of answer scores and cannot limit the specific number of answers to the question, so the user has to determine the number of answers by guessing. As shown in Fig. 1, the question "who held the position of secretary of state when Andrew Jackson was

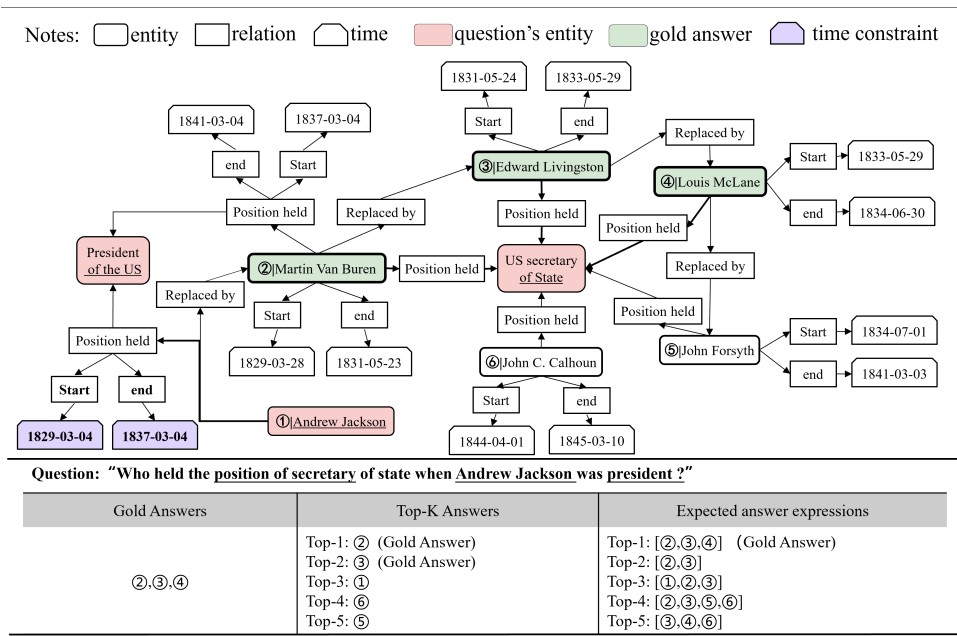

Notes: ☐ entity  ☐ relation  ⬡ time  🟥 question's entity  🟩 gold answer  ⬟ time constraint

Question: "Who held the position of secretary of state when Andrew Jackson was president ?"

| Gold Answers | Top-K Answers | Expected answer expressions |
|---|---|---|
| ②,③,④ | Top-1: ② (Gold Answer)<br>Top-2: ③ (Gold Answer)<br>Top-3: ①<br>Top-4: ⑥<br>Top-5: ⑤ | Top-1: [②,③,④] (Gold Answer)<br>Top-2: [②,③]<br>Top-3: [①,②,③]<br>Top-4: [②,③,⑤,⑥]<br>Top-5: [③,④,⑥] |

**Figure 1   An expression of answers to the question and an excerpt from the Wikimap of the question.**

president?" has three accurate answers, "Martin Van Buren, Edward Livingston, and Louis McLane." In the traditional answer representation mode, users can only get a few answers with high scores according to the Top-K list, but they cannot be sure about the specific number of answers that meet the semantic conditions.

**Answers with higher scores are not necessarily correct.** There is a special case where a specific number of answers to a question has been given, but there are still wrong answers among the candidates. Therefore, in general cases, there are still false positives for answers with high scores. In the list of the top five answers in Fig. 1, only the first two are standard answers, the answer with the third high score is wrong, and the third accurate answer is not obtained by reasoning, so there are still errors in the answer combination screened by the user's intuition.

**Time constraints are not fully considered in multi-answer temporal problems.** The WikiData data excerpt for the question in Fig. 1 shows that Andrew Jackson was president of the United States for a period of time (1829-03-04, 1837-03-04), and three secretaries of state met this time constraint. Other candidates for secretary of state should be eliminated because they do not meet the time constraint. Most knowledge graph-based question answering (KGQA) models however ignore the important role of timing constraints when dealing with multi-answer questions, leading to incorrect results. The key to answering such multi-answer temporal questions is to determine the candidates that satisfy the time constraint interval of the answer. A time fact can be considered as a correct answer only if it conforms to the temporal logic of the problem, that is, the temporal constraint represented by a given explicit or implicit fact needs to be satisfied.

This article therefore proposes a Multi-Answers Temporal Question Answering (MATQA) component for multi-answer reasoning, which can be combined with any KGQA system to improve the answering effect. The time constraint on the correct fact in the knowledge graph (KG) candidates makes it possible to output all the standard answers. To address the above problems, MATQA proposes the following solutions. First, inspired by the multi-paragraph open-domain question answering, after introducing the multi-answer question into the field of knowledge graph temporal question answering, the revalidation framework is used to improve the existing Top-k answer display form, and the question answering process with a certain number of answers is constructed. Second, the correct initial answers among the candidate answers are filtered by embedding the question and answer pairs into the graph as boot nodes. Finally, since multiple answers to a question may have the same type or relationship, and answers to timing questions may have the same time constraints, this article filters answers from two aspects: semantic constraints and time constraints. Our goal is to select answers that are also close in terms of semantics and time interval.

At the same time, the incorrect answers with high scores can be filtered again at the semantic level to ensure the accuracy. Experiments using a recent temporal question answering benchmark and a set of competitors based on unstructured text sources show the advantages of MATQA: The model can give the number of correct answers based on the knowledge graph, and can use the time information of the temporal question to filter the answers. Given a new answer expression, it can better guarantee the quantity and quality of the answers.

In summary, the key contributions are three-fold:

- Multi-answer reasoning is introduced into temporal knowledge graph question answering to improve Top-k, and a new answer expression is proposed, which gives the user the exact number of answers.
- Based on the revalidation framework, a component that contains time information is designed to guarantee the quantity and quality of answers.
- New evaluation indicators $P@1^m$ and $Hits@5^m$ for multiple answers were designed, and a series of experiments were conducted based on these indicators. Experiment result shows that MATQA can not only infer the number of answers to temporal questions, but also take into account the accuracy of knowledge question answering.

## RELATED WORK

**Top-k algorithm.** The traditional Top-k method aims to return the top k answers that are closest to the expected value. The main idea is to filter a series of candidate matches constructed according to the similarity criterion so as to obtain the answer that matches the target value. Each step of KGQA, such as named entity recognition, entity disambiguation, and entity linking, results in a ranked Top-k list. The whole question answering process is the Top-k retrieval of multi-link ranking mechanism fusion. The main methods are Fagin algorithm and threshold algorithm, and the core task is to sort the candidates of multiple dimensions, and then calculate according to a specific pruning strategy (*Auer et*

*al., 2008*). For example, *Christmann, Roy & Weikum (2022)* fused the quantitative scores such as semantic coherence of candidate items, connectivity of knowledge graph, relevance to the question, *etc.,* to reduce the candidate domain in knowledge question answering, and then used the threshold algorithm to filter the score list of multiple indicators to obtain the most relevant candidate neighborhood to the question. *Wang et al. (2021)* filtered the semantically weighted scores of edges using upper and lower bound filtering and defined a star Top-k query scheme with early termination of matching. Top-k query is related to the quality of answers. However, the traditional Top-k query is presented in the form of a single answer list, which cannot reflect the standard answers of multi-answer questions, including the number and accuracy of answers. MATQA extends the single-answer display form to a multi-answer one, which can better ensure the quality in multi-answer question answering.

**Multi-answer question retrieval based on unstructured text sources.** Unstructured text sources often organize knowledge in the form of articles or paragraphs and are crucial in the field of question answering. In practice, multiple-answer questions play an important role in various assessment methods (*Maheen et al., 2022*). Open-domain question answering based on multi-paragraph multi-answer reasoning challenges the ability to comprehensively utilize evidence from large-scale corpora. Due to the ambiguity and openness of questions, a question often has multiple correct answers. Predicting the answer contained in each paragraph in turn after retrieving the reordered paragraphs has become the mainstream question answering paradigm in this field. Pre-trained models are widely used in question and answer systems (*Ahmed et al., 2023*), for example, AMBIGNQ (*Min et al., 2020*) uses BERT model to sort paragraphs and generate answers in turn. *Shao & Huang (2022)* proposed the "recall and revalidation" framework to separate the reasoning process of each answer and used the new evidence obtained from recall to verify the answer. Although unstructured multi-answer question answering has received extensive attention, the multi-answer question answering based on structured data cannot meet the needs of obtaining all correct answers to the question. Therefore, it is of great practical significance to extend multi-answer question to knowledge graph question answering.

**Multi-answer reasoning based on temporal knowledge questions.** Good progress has been made in the question answering of temporal questions. A series of advanced schemes (*Jia et al., 2021*; *Saxena, Chakrabarti & Talukdar, 2021*; *Mavromatis et al., 2022*; *Jiao et al., 2022*; *Chen et al., 2021*) have proved that the processing of time information in the question is helpful to guarantee the quality of complex knowledge question answering. The time information contained in the question limits the time interval of the answer. When the semantic constraints are satisfied, the number and accuracy of the answers to the multi-answer question are measured by the time interval. The facts beyond the time interval do not satisfy the user intention and should be excluded from the answer output. As a special branch of temporal questions, the multi-answer question faces great challenges. The single answer list and false positive answers make it difficult for users to determine the number and accuracy of answers to a question. This paper therefore aims to expand the answer expression form of multi-answer temporal question, and investigate the factors

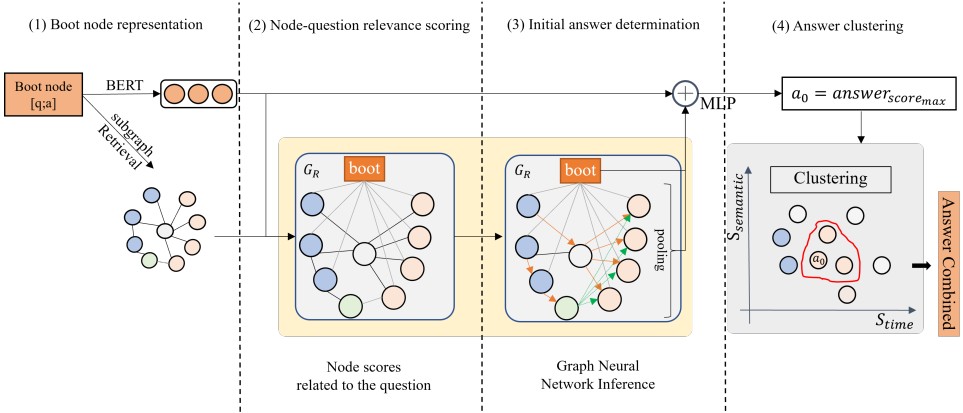

**Figure 2** **The structure of MATQA.** The component can be attached to the question answering system. Based on the revalidation framework, it uses the boot node (another form of Q&A pair) representation, as well as the KG node score related to the question, to determine the initial answer, and finally obtains the answers through the time and semantic dimension of the alternative answer clustering.

that ensure the quality of temporal question answering based on the complete question answering process.

# RESEARCH METHOD

**Task description**: The objective of this paper is to answer multi-answer temporal questions with question answering pair information and structured knowledge. For a given question $q$ and its candidate answers set $\mathcal{A}$, MATQA aims to determine the number of valid answers to question $q$ and identify correct entities or attributes within the candidate answer set $\mathcal{A}$. **Approach Introduction:** Fig. 2 presents the overall structure of MATQA. It uses four modules to perform the process of answering multi-answer temporal questions, corresponding to the **boot node representation** module, **node-question relevance scoring** module, **initial answer determination** module and **answer clustering** module. First, in the boot node representation module, the Q&A pair is associated with the knowledge graph as a special node we call *boot* node, which can bridge the information gap between Q&A pair and subgraph in the subsequent reasoning process, and guide the model to approach the standard Q&A. Second, the node-question relevance scoring module is used to calculate the relevance score between the key entities in the resolved triplet facts in the question and the boot node and retrieve a subgraph consisting of the *KG node* (nodes in the knowledge graph, including entities and attributes) most relevant to the question based on the relevance score. Subsequently, the initial answer determination module aggregates and updates the information of the boot node and the subgraphs through the attention-based graph neural network (GNN), and the possible answers with the highest score is deduced. Finally, the answer clustering module clusters all candidate answers through the time constraints parsed from the question, and uses the clustering results as the final answer set to the question.

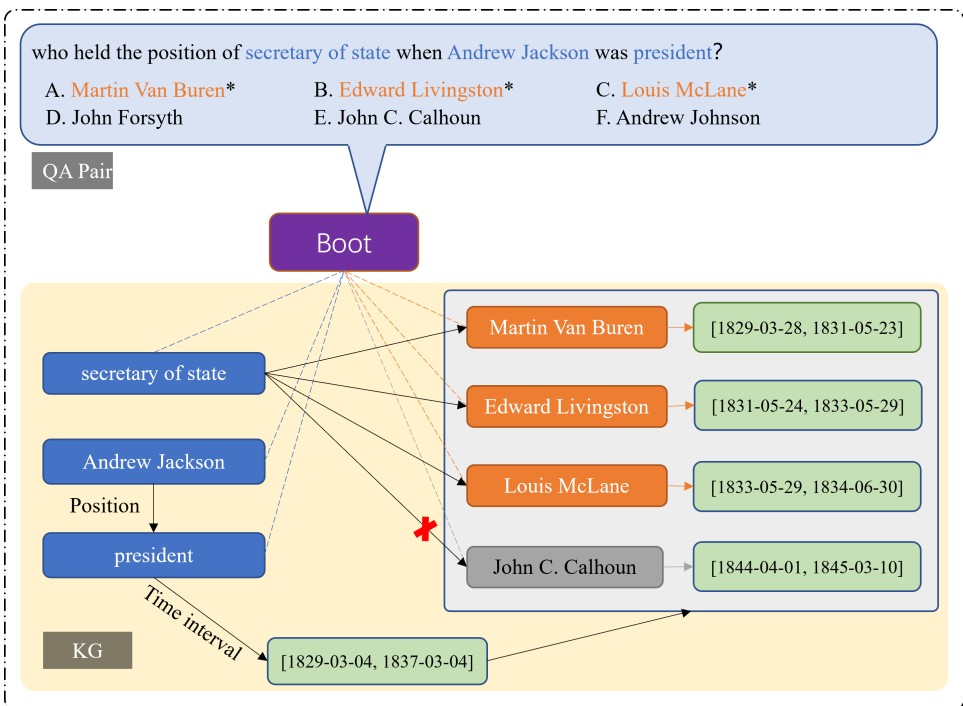

**Figure 3** **Diagram of "inference graph".** The orange dotted line points to the entity related to the answer, and the blue dotted line points to the entity in the question. Through time constraints, it can be inferred that John C does not meet the conditions.

## Boot node representation

In order to use the answer information to guide the question reasoning, the question $q$ and the candidate answer set $\mathcal{A}$ provided by other question answering schemes are together inserted into the knowledge graph as a special node, known as boot node (*boot*), denoted as $[q; a]$, as shown in Fig. 3. Herein, $\mathcal{A}$ can be a traditional form of Top-k solution to question $q$ given by any question answering scheme, and the standard answer in the candidate solution set $\mathcal{A}$ is clearly marked. In the special nodes formed by Q&A pairs, the question is taken as the starting point of the reasoning model, and the answer as the end point, implicitly expressing the information of the question and answer context. The boot node is associated with entities contained in the question, and the mapping item of the boot node and the marked standard answer node in the knowledge graph is linked, and the new relation "gold answer" is given, which is shown by the orange dotted line in Fig. 3. Therefore, a new answer-guided knowledge graph is constructed between the boot node and the knowledge graph, and between the answer node and the corresponding boot node, known as inference graph $G_R$ herein.

The boot node is regarded as a long sequence text and encoded by BERT, where $f_e$ is the encoding function.

$$\boldsymbol{boot}^{BERT} = f_e(text(boot)). \tag{1}$$

After the boot node is given, the subgraph $G_{sub}^{boot} = (v_{sub}^{boot}, e_{sub}^{boot})$ after entity link is extracted from knowledge graph $G = (V, E)$, where $V$ is the set of entity node of the knowledge graph, $E$ is the set of relationships between entity nodes, $v_{sub}^{boot}$ is the entity nodes in all boot nodes extracted from the graph, $e_{sub}^{boot}$ is the relationship nodes in all the boot nodes extracted from the graph, and $G_{sub}^{boot}$ is the subgraph associated with the boot node extracted from the knowledge graph.

## Node-question relevance scoring

There are many paths unrelated to the question in the subgraph after entity link disambiguation. As shown in Fig. 1, Martin Van Buren's path as president is unrelated to his path as Secretary of State. These unrelated paths cause the model to waste a lot of time in the inference process to exclude invalid paths. To address this problem, this paper uses the question correlation fact determination module to calculate the similarity score between the boot node and KG fact node.

$$S_{sub}^{boot} = f_h(f_e([text(boot); text(v_{sub}^{boot})])), \tag{2}$$

where $f_h$ is a function to obtain the head of the language model (here it is used to obtain the head of BERT), and $f_h(f_e())$ is the probability that the boot node is connected to the subgraph node; $S_{sub}^{boot}$ is the score of correlation between the boot node and the subgraph node, which describes the importance of each node to the boot node, and is used to prune the inference graph $G_R$.

## Initial answer determination

The answer with the highest score in the question answering system has the greatest probability of being the standard answer. This paper therefore finds out the most likely answer to the multi-answer question through subgraph reasoning, and regards it as the correct answer. MATQA's reasoning process is based on the graph attention GAT framework.

In an $l$-layer graph network model, for a node $v \in V_{sub}$ in any subgraph, vector initialization is performed by BERT encoding, *i.e.*, $h_v^0 = f_e(text(v))$. Then the updating model can be expressed as:

$$h_v^{l+1} = \left( \sum_{n \in N_v \cup \{v\}} \delta_{nv} m_{nv} \right) + h_v^l, \tag{3}$$

where $h_v^{l+1} \in \mathbb{R}^D$ is the representation of node $v \in V$ (in the form of a D-dimensional vector), $N_v$ is the set of neighbors of node $v$, $m_{nv}$ is the message from each neighbor node $n$ to node $v$, and $\delta_{nv}$ is the weight of the message from node $n$ to node $v$. The calculation of message $m_{nv}$ should take into account the characteristic $h_n^l$, type $u_n$, and time attribute $t_n$ of the node, as well as the embedded relation $r_{nv}$. The calculation formula is as follows:

$$m_{nv} = Linear(h_n^l, u_n, t_n, r_{nv}), \tag{4}$$

where $u_n$ is the type's one-hot encoding of the neighbor $n$ of node $v$, $t_n$ is the embedded time attribute of neighbor node $n$, and $r_{nv}$ is the embedded relation between nodes $n$ to $v$.

To calculate the attention weight vector of nodes $n$ to $v$, query vector $\boldsymbol{q}$ and key vector $\boldsymbol{k}$ are constructed according to node types:

$$\begin{aligned} \boldsymbol{q}_n &= Linear(\boldsymbol{h}_n^l, \boldsymbol{u}_n, \boldsymbol{S}_n^{boot}) \\ \boldsymbol{k}_v &= Linear(\boldsymbol{h}_v^l, \boldsymbol{u}_v, \boldsymbol{S}_v^{boot}, \boldsymbol{r}_{nv}) \end{aligned}, \tag{5}$$

where $Linear$ is a linear transformation that converts the input into a D-dimensional vector. $\boldsymbol{S}_n^{boot}$ and $\boldsymbol{S}_v^{boot}$ is the correlation score between the boot node and nodes $n$ and $v$. The final attention weight vector can be obtained by formula (Eq. 6) below.

$$\delta_{nv} = \frac{exp(\gamma_{nv})}{\sum_{v' \in N_n \cup \{n\}} exp(\gamma_{nv'})}, \quad \gamma_{nv} = \frac{\boldsymbol{q}_n^T \boldsymbol{k}_v}{\sqrt{D}}. \tag{6}$$

Then the reasoning process of the initial answer $p(a_0^i|q)$ is given by:

$$p(a_0|q_i) = exp(MLP(\boldsymbol{boot}^{BERT}, \boldsymbol{h}_{boot}^l, G_{sub}^{pooling})), \tag{7}$$

where $\boldsymbol{boot}^{BERT}$ is the vector representation of boot node, $\boldsymbol{h}_{boot}^l$ is the updating representation of the boot node at the $l$-th layer, and $G_{sub}^{pooling}$ is the pooling representation of subgraph.

## Answer clustering

After the initial answer $a_0$ is obtained, the rest of the answers to the question should be deduced. Since all answers to the question should meet the same constraints, including the semantic and time constraints, MATQA processes the other answers through clustering. In order to correctly measure the gap between the alternative answer and the initial answer $a_0$, the subgraph path $(V_{sub}, E_{sub}, a_{other})$ of the alternative answer is extracted to calculate the semantic similarity score between it and the path $(V_{sub}, E_{sub}, a_0)$ of the initial answer.

$$S_{semantic} = \cos[(V_{sub}, E_{sub}, a_{other}), (V_{sub}, E_{sub}, a_0)]. \tag{8}$$

The final answer to each question is constrained by the time interval. Therefore, the matching between the time interval of the fact and the real time interval of the question can exclude the answer that does not satisfy the condition. KG retrieval and TimeML (*Pustejovsky et al., 2003*) are used to calculate the time constraint interval of the question, which is $[T_s, T_e]$ ($T_s$ and $T_e$ are the start time and end time, respectively). At the same time, the time interval $[T_s^{other}, T_e^{other}]$ of the fact corresponding to the alternative answer is extracted. The final predicted score of time similarity $S_{time}$ can be obtained by:

$$S_{time} = \text{ReLU}\begin{cases} 1, T_s < T_s^{other} \text{ and } T_e^{other} < T_e \\ -1, T_s^{other} < T_s \text{ or } T_e^{other} > T_e \end{cases}. \tag{9}$$

We use the K-means algorithm with $K = 2$ for clustering, where one cluster center is set to the initial answer $a_0$. Setting K to 2 is because we believe that the correct answers will be closely distributed in the neighborhood of the top-1 answers in the initial answer list in the entire answer candidate solution space. Therefore, when the number of clusters is set to 2, the correct answer combinations can be aggregated into one cluster, while the remaining answers will be classified into another cluster.

The ReLU function is commonly used as an activation function, but this article uses its rectifying properties to filter $S_{time}$: when the answer does not meet the time constraints, the score is truncated to 0 through ReLU. The answers that satisfy the semantic and time constraints after clustering are regarded as the true predicted answers $\mathcal{A}$ to the question $q$. Each row of Top-k is a combination of answers, as shown in the expected answer expressions in Fig. 1.

## EXPERIMENT

### Datasets

TimeQuestions (*Jia et al., 2021*) is a wikidata-based question-answering data set consisting of 16,181 Q&A pairs, among which 9,708 questions are used for training, 3,236 for verification and 3,237 for testing. The type of each question (explicit, implicit, time, and order) is indicated in the Q&A pairs. At the same time, the signal words for time interaction in the question are specified, such as before/after, start/end, *etc*. In order to process the multi-answer questions, all question pairs with more than one answer are extracted from the TimeQuestions data set to construct the multi-answer TimeQuestions data set. The new multi-answer question dataset contains 2,264 training sets, 778 verification sets and 801 test sets, and the labels of the question type and time signal.

### Evaluation metrics

Two measures are used to evaluate the quality of answers to the multi-answer question.

- $P@1^m$ (the precision of multi-answers): For a new answer form given in a question, the highest-ranked combination of answers has a precision of 1 when the combination is exactly the same as the standard answers (both in the quantity and the label), which is denoted as $P@1^m_{hard}$. When the highest-ranked answer combination contains all the standard answers, that is, the first result of the prediction includes other results besides the standard answers, it is denoted as $P@1^m_{soft}$ with broader constraints.

- $Hits@5^m$ (the hits of multi-answers): The combination of answers depends on the number and label of answers. The label needs to satisfy the semantic matching relation of the question, and the number is all possible solutions that satisfy the semantic constraints. Because of the complexity of language questions, semantic constraints cannot be fully satisfied, and there are many possible combinations of answers. Under the new answer expression form, the first five groups of answers are ranked in descending order of the proportion of the standard answers on the list. If a list containing any subset of the standard answer appears in the first five positions, it is set to 1, otherwise to 0.

### Baselines

The goal of the traditional Top-k based QA system on multi-answer questions is to predict every answer that may belong to the correct answer combination, while MATQA, as a plug-in component of the QA system, converts the goal of the QA system into directly predicting answer combinations. The system's answer output has been completely changed so that the prediction results are presented in the form of a list of answer combinations. For this reason, the experiment in this section aims to reflect the effectiveness of this

**Table 1  Comparison of results of MATQA.**

| Model | $P@1^m_{hard}$ | $P@1^m_{soft}$ | $Hits@5^m$ |
|---|---|---|---|
| TransE+MATQA | 0.402 | 0.439 | 0.513 |
| EXAQT+MATQA | 0.431 | 0.453 | 0.546 |
| TERQA+MATQA | 0.459 | 0.472 | 0.538 |

method on multi-answer questions through the metrics $P@1^m$ and $Hits@5^m$ designed for this predicted answer form, rather than verifying the superiority of this method compared to other methods.

- TransE: it is the most classical vector embedding method which completes the missing answers according to the translational semantic invariance law.
- EXAQT (*Jia et al., 2021*): it is an end-to-end temporal question answering scheme, which for the first time builds the temporal question answering system on wikidata, a large-scale open-domain knowledge graph. It does not require the process of constructing a temporal knowledge graph. The final answer prediction and accuracy is performed using a relational graph convolution network (R-GCN) by augmenting the embedding of subgraphs and questions, performing temporal augmentation of subgraphs, or reconstructing subgraphs to augment recall in three ways.
- TERQA (*Yao et al., 2022*): On the basis of EXAQT, inspired by capsule network, TERQA improved the fusion of time features and triplet features and learned the exact dependence between time features and triplet facts, which enhanced the accuracy of the model to predict the answer.

## Experimental settings

MATQA uses PyTorch for implementation, and sets the vector embedding dimension after BERT initialization to 200. It has five layers of GNN, each of which with a dropout of 0.2. Moreover, it uses Adam for initial answer inference optimization and ReLU as a filter on time constraint scores. Furthermore, batch_size is set to 32, learning rate to 2e−3, and cluster number to 2.

## RESULTS

### Key findings

Table 1 shows the effects of multi-answer judgment on the multi-answer question data set. The index $P@1^m_{hard}$ demonstrates that MATQA can improve the traditional Top-k expression form to make each line a new form of a list of answers, which is consistent with the expected human expression form in Fig. 1. Therefore, MATQA can better meet user's requirements on the number and accuracy of questions with multiple answers. At the same time, MATQA has proved that its effectiveness is largely related to the alternative answers provided. That is, the more accurate the candidate answers, the more accurate the initial answer, and the better the final result after clustering.

Through the revalidation framework of "initial answer → clustering", MATQA can provide a solution to the multi-answer temporal reasoning question. The primary

**Table 2  Results of TransE + MATQA after removal of module.**

|  | Model | $P@1^m_{hard}$ | $P@1^m_{soft}$ |
|---|---|---|---|
| No boot nodes |  | 0.382 | 0.391 |
| GNN | No node types | 0.398 | 0.401 |
|  | No score of nodes related to question | 0.386 | 0.394 |
|  | No pooling layer | 0.382 | 0.389 |
| Clustering | No semantic constraints | 0.254 | 0.287 |
|  | No time constraints | 0.305 | 0.348 |

shortcoming of MATQA is that its final output is largely affected by the initial result. In other words, in the case of an incorrect initial answer, the subsequent clustering module cannot correct it and can only make invalid predictions on a wrong basis.

## Disambiguation experiment

Table 2 shows the results of MATQA after removing each module. It can be seen that the introduction of the boot node enables the question and the candidate answers to inspire the inference model. In addition, the boot nodes have positive feedback to $P@1^m$. In the case of no boot nodes, the $P@1^m$ score is the lowest relative to the case with a boot node, which means the QA model cannot get the information guidance of hidden answer, and the Q&A context cannot be updated with KG, which cannot bridge the information gap between question and knowledge graph and thus damages the system performance ($P@1^m_{hard}$:40.2% → 38.2%, $P@1^m_{soft}$:43.9% → 39.1%).

When semantic constraints are removed during clustering, the model effect declines most seriously, because the clustering of answers mainly measures the degree of fact similarity. Additionally, among temporal questions, a large proportion have answers within a specific time constraint interval. When time constraint is removed, the entities of the answers cannot be measured by time constraint, which will easily lead to incorrect answers. Finally, the addition of the boot node makes up the information gap between the question context and the knowledge graph, and has a great influence on the determination of the initial answer. Removing modules from GNN also has an effect on the prediction of the final initial answer.

## Typical questions

The effectiveness of MATQA is fully demonstrated by three typical questions. In Table 3, the question Q1 has the standard answers of "Super Bowl 'IX', 'X', 'XIII', 'XIV', 'XL', 'XLIII'". The model has accurately predicted the number of answers and the correct answer. In the traditional Top-K method, it is difficult to obtain the correct answer combination due to the difficult K setting. For example, when the setting of k is less than the number of correct answers, it will result in the output of incomplete answers. Taking Q1 in Table 3 as an example, when $k = 3$, the three answers with the highest scores will be output, namely: IX, X, XIII , the correct answer with the lower score is lost. For another example, in the answer list obtained based on the top-k method, there may be cases where the correct answer is not the highest-scoring answer. Even if the correct k value is selected,

**Table 3   Top-1 results of improved questions.**

| Question | Gold answers | Predicted answers |
|---|---|---|
| Q1: In which year, did the Steelers win the super bowl, the latest occasion? | Super Bowl 'IX','X', 'XIII', 'XIV', 'XL', 'XLIII' | Super Bowl 'IX', 'X', 'XIII', 'XIV', 'XL', 'XLIII' |
| Q2: Who ran against Lincoln in the 1864 presidential election? | "John C. Breckinridge" and "Stephen A. Douglas" | "John C. Breckinridge" and "Stephen A. Douglas" |
| Q3: When did owner Fred Wilson's sports team win the pennant? | "1969 World Series" and "1986 World Series" | "1969 World Series" and "1986 World Series" |

**Table 4   Incorrect results obtained by MATQA.**

| Question | Gold answers | Predicted answers |
|---|---|---|
| Q1: What is inflation rate of Dominica that is point in time is 1983-1-1? | "2.7" | "ACM Software System Award" and "Turing Award" |
| Q2: When did Anne Hathaway begin attending New York University and when did she graduate? | "1995" and "1998" | "History of art" |

it may lead to the wrong selection of the final combined answer. It is proved that MATQA framework has a good effect on the processing of multi-answer temporal questions, and makes up the defects of traditional top-k which cannot show the number of answers and has false positive results.

## Error types

As shown in Table 4, we selected two questions Q1 and Q2 with numerical answer types as cases of incorrect answers for analysis. question Q1 expected a numeric answer of 2.7, but instead returned multiple unrelated entities as the answer. This shows that MATQA still cannot accurately determine the number of answers through semantic and time constraints for some single-answer questions, and there is room for further improvement. question Q2 expected to get two numerical answers "1995" and "1998", but actually got a single entity as the answer. We believe that this phenomenon may be related to the initial answer generation of the upstream task. As we describe in Section: Initial result determination, MATQA will use GNN to perform inference on the extracted subgraph to obtain a preliminary answer. The effect of inference on the subgraph depends on the extent to which GNN can accurately learn the nodes. feature. For entity-type answers, each entity can have multiple neighbors. The rich neighborhood structure allows GNN to capture the characteristics of entities very well. However, for numerical nodes, most of them are only used for directly related nodes. For example, the expected answer of Q2 is "2.7". This value is quite special and it is difficult to find a second node that refers to this value. Therefore, the GNN is likely to make errors in capturing its features, which in turn leads to the wrong exclusion of the answer, so the downstream The clustering module will not be able to get the correct answer.

## CONCLUSION

In this study, MATQA defines the true number of answers and eliminates false positives through a "revalidation" framework. The combined use of initial answer establishment and semantic time based dual factor clustering ideas was shown to have a positive effect on the number of answers and correctness of questions. Previous research (*Rubin et al., 2022*) has shown that the revalidation framework is able to take full advantage of the information collected to further filter the answers. This is consistent with the study in this paper. Further, the "revalidation" framework was shown to be able to determine not only the correctness of answers but also the number of answers, with only the addition of semantic and temporal constraints on clustering. Based on this, this paper shows that the "revalidation" framework in the form of "initial answer → clustering" can provide a solution to the multiple answer reasoning problem in the context of temporal knowledge quiz. Experimental results on a large number of complex multi-answer temporal questions show that MATQA can improve the most advanced general Top-k question answering scheme. However, MATQA suffers from severe upstream error-dependent transmission. When the initial answer is wrong, the subsequent clustering module cannot correct the result, but only makes invalid predictions based on the original one.

Despite its drawbacks, this study provides a solution to multi-answer questions in a structured temporal knowledge Q&A scenario and points out that the key to multi-answer questions lies in the number of answers and false positive result filtering. Meanwhile, the introduction of bootstrap nodes enables questions and candidate answers to shed light on the inference model, and subsequent updates jointly utilize bootstrap nodes and subgraph domains to bridge the information gap between questions and knowledge graphs. Based on the existing research, the establishment of initial answers and the refinement of clustering factors will be the next step of research to be considered.

## ACKNOWLEDGEMENTS

We thank Zhihong Shao, Zhen Jia, and Michihiro Yasunaga for their achievements that inspired this work, and we would like to thank the anonymous reviewers for their helpful remarks.

### Funding
The authors received no funding for this work.

### Competing Interests
The authors declare there are no competing interests.

### Author Contributions
- Junping Yao conceived and designed the experiments, authored or reviewed drafts of the article, and approved the final draft.

- Cong Yuan conceived and designed the experiments, performed the experiments, analyzed the data, performed the computation work, prepared figures and/or tables, and approved the final draft.
- Xiaojun Li conceived and designed the experiments, authored or reviewed drafts of the article, and approved the final draft.
- Yijing Wang conceived and designed the experiments, performed the experiments, analyzed the data, performed the computation work, prepared figures and/or tables, and approved the final draft.
- Yi Su analyzed the data, authored or reviewed drafts of the article, review and verify, and approved the final draft.

## Data Availability

The raw measurements are available in the Supplementary File.

## Supplemental Information

Supplemental information for this article can be found online at http://dx.doi.org/10.7717/peerj-cs.1725#supplemental-information.

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
