# Peer review of "Beyond top-k: knowledge reasoning for multi-answer temporal questions based on revalidation framework"

_PeerJ Computer Science, doi:10.7717/peerj-cs.1725_

## Round 0.1 · original submission · Major Revisions

Please address all the concerns identified in the reviews. Prominent among these are the quality of the presentation itself, but also the extent and nature of benchmark experiments.

·

Basic reporting

This paper proposes a new method to answer the knowledge-based multi-answer temporal questions which is built on a reverification framework. This is an interesting problem in the area of knowledge retrieval/question answering/multi-answer reasoning from unstructured text. In this paper, given a question that is related to a temporal aspect, the authors assume the pre-existence of the initial candidates of answers. Then, this paper proposes a reverification component to suggest the most valid (i.e., top-k) answer combination from the candidates, which includes four sub-components, namely, a Bert encoding layer for the initial input, a subgraph retrieval and node scoring layer, a initial result determining layer applying the graph attention network, and a refinement layer to prune some irrelevant answers using the clustering technique.

The strong points of this paper:
S1. This paper tackles a novel and challenging problem of multi-answer reasoning of temporal questions.
S2. This paper gives a well-structured background introduction of related literature, and a clear statement of the challenges with the existing approaches (e.g., undermined number of answers, high false positive results and time constraints not considered), which are targeted and addressed in this paper using an effective approach with four modules.
S3. Ablation tests were conducted to verify the effectiveness of each module in the proposed method.

The weak points of this paper (some details could be found below) (more prioritized appearing first):
W1. The results may be more convincing if the baselines with existing approaches only are also compared (see the "Experimental design" part for details).
W2. It is not clear how the proposed method could obtain the correct number of answers in the method design. The authors may wish to add some discussion or intuition about that.
W3. The description of the method seems to have some non-explicit/unexplained terms and definitions (see below for details).
W4. Some figures in this paper lack clear explanation to be understood, and/or it is difficult to link the figures to the text in this paper (see below for details).

Detailed comments related to this part are shown below.
D1. The presentation of this paper could be improved with more explicit terms and definitions so that the ideas and method could be presented more clearly (as mentioned in W3). Some notable points to be improved are list below.
1). Please give the clear form (possibly with examples) of q_i and a_m^i in the task description in line 156-159. Here, it is not clear to me why there are m *sets* of candidate answers and whether a_m^i denotes all those *sets*.
2). Term KG node is not defined while it first appears in line 162.
3). The connection between the four named modules in line 161-163 and the paragraph following this part (which I guess should elaborate the four modules) seems weak.
4). The notation f_h \circ f_e in line 192 is not explicitly matched to Equation (2), and f_h is not clearly defined.
5). Across the description in page 6, there are some undefined notations. Examples are function linear (and Linear), q_n, k_v, S^{boot}_{n \in sub}, and D.
D2. The figures in this paper need better explanation (as mentioned in W4).
1). In Figure 1, although the question and answers in the lower part show a good example, the diagram in the upper part has too much information which is confused to me, where the different shapes of boxes are hard to tell, and it is not clear what different colors and different connectors refer to.
2). In Figure 2, it is not easy to match each module in the proposed method to the part in the figure. Moreover, the details in this figure are not explained, e.g., the different colors and the changes in the subgraphs, how the clustering works by the example shown in this figure.
3). In Figure 3, similarly, most parts are not explained about how the inference graph exactly works.
D3. Although most of the language/technical text used in this paper is correct, there are some minor issues to improve. Examples are as follows.
1). Add a space before each citation.
2). In the caption of Figure 1: "A expressions of" -> "An expression of".
3). In line 192 (and other places similarly): "Where" -> "where".
4). Under Figure 3: "a l layer" -> "an l-layer".
5). A grammatical issue in line 200.
6). In line 203: "at l layer" -> "at the l-th layer".
7). In line 215: the citation [12] should follow a correct citation form.
8). In Table 3&4: add a horizontal line to split each example.
9). In line 187 (and other places similarly): we usually say V is the *set* of all nodes in a graph, while the text here is ambigious.
10). Equation (6) possibly has some issues.
11). Missing full stop in line 288.

I have checked other aspects (e.g., raw data), and I found no other issues related to this part except what I have mentioned above.

Overall, I believe that this paper could be substantially improved after addressing the presentation issues elaborated in my comments in this part. Moreover, some additional experiments could be added to better verify the effectiveness of the proposed method, which is elaborated in the next part.

Experimental design

This paper targets a well-defined research problem of multi-answer reasoning of temporal questions, which addresses the gaps/challenges in the literature. The authors propose a novel structure to tackle the problem which makes a clear contribution.

Generally, the experimental design in this paper is well replicable and can verify the effectiveness of the proposed method to some extent with examples of case study. However, one possible issue is that the compared baselines do not involve a method that only use existing techniques. I understand that the proposed structure is attached to the traditional QA model, but the experimental results should show the improvement of using the proposed structure (e.g., TransE+MATQA) over the existing method only (e.g., merely TransE). In addition, the authors are suggested to consider other baseline to replace MATQA in the framework to show the superiority of MATQA (e.g., TransE+XX, where XX could be a baseline method that use some traditional top-k technique to obtain results from candidates or even a trivial baseline).

Validity of the findings

The provided data and code seem to be complete and can support the technical content and results shown in the paper. The findings and results basically addresses the research problem and support the conclusion to some extent. However, as mentioned in the previous part, improvements are required to strengthen the validity of this paper.

·

Basic reporting

--Although the scientific content of the paper seems good for publication, the structure of sentences needs some revision. Simpler, shorter sentences should be preferred to easily convey the message to the reader. Some examples are mentioned in other comments below and also highlighted in the attached pdf of the paper.
--The sentence on Line 40-43, “Retrieval and reading…” is too long. It can be split into two shorter sentences for easy reading. Same with Line 43-45, “For example…”. And Line 95-99, “Finally,..”
-- Line 98 should remove word "that", as in “…high scores are eliminated.”
--Provide full-forms for all acronyms used. These full-forms can be either in the text or as a separate section at the end of the paper. For example, Line 250 R-GCN,(Relational Graph Convolution Network), and Line 258 GNN (Graph Neural Network). There are many instances where the full-form is missing.
--A thorough literature review for previous work that classifies similar answering schemes is provided. However, the format for referencing previous literature is incorrect in multiple places as highlighted in the attach pdf. For example, it mentions, (Liu et al., 2021) on line 47 but it should be written as “Liu et al. (2021)”. In fact, some places mention a number in square brackets but the reference section is not organized numerically. This needs to be revised or clarified on Line 29, 45, 48, 51, 54, 120, 137, 215, 247, 313
--Two papers published recently in PeerJ may be of significance to this paper and should be added to the literature review:
• Maheen, F., Asif, M., Ahmad, H., Ahmad, S., Alturise, F., Asiry, O., & Ghadi, Y. Y. (2022). Automatic computer science domain multiple-choice questions generation based on informative sentences. PeerJ Computer Science, 8, e1010.
• Ahmed, M., Khan, H., Iqbal, T., Alarfaj, F. K., Alomair, A., & Almusallam, N. (2023). On solving textual ambiguities and semantic vagueness in MRC based question answering using generative pre-trained transformers. PeerJ Computer Science, 9, e1422.
--The syntax for mentioning the time period can be written in a better way. For example, on Line 80, instead of writing 18290304, it can be stated as 1829-03-04 for easy reading and comprehension. A similar approach can be implemented in the figures where the time period is stated.
--In Line 127, what does “answer of answers” mean? This should be clarified in the text.
--In Line 245, insert the word “which” as in, “….embedding method WHICH completes….”
--In Line 253, insert “TERQA” as in, “… inspired by the capsule network, TERQA improved..” And check spelling of enhanced on Line 255.
--Line 272, instead of “out”, it should say “output”
--Line 273, remove the word "the" and add "s" after prediction, as in “….cannot correct it and can only make invalid predictionS on a wrong basis.”

Experimental design

--Line 157-159, “Given a problem…”. This sentence can be split into two shorter sentences to explain it better. What does a_m^i signify?
--Line 199, “In a l layer graph network model….., vector initialization is performed it by …”. Remove “it”.
--Line 205, why is the number of clusters set to 2?
--In Line 259, Relu is used for time constraint score optimization. First, the authors need to clarify if Relu was used for optimization (like ADAM) or for activation. It is usually used for activation layer in neural network. Some application studies suggest other optimizers like the Parametric Relu which outperform the classical Relu in multi-classification problems (example: Sandhu, H. K., Bodda, S. S., & Gupta, A. (2023). Post-hazard condition assessment of nuclear piping-equipment systems: Novel approach to feature extraction and deep learning. International Journal of Pressure Vessels and Piping, 201, 104849.). Were any other activation functions used to assess the best possible solution?
--The title of the paper mentions “reverification” but the paper highlights a “revalidation” framework at multiple places. The authors should clarify whether the proposed methodology is implementing a “reverification” or a “revalidation” framework. In scientific or experimental terms, these words do not mean the same thing.

Validity of the findings

--More explanation is needed to justify the results of Table 1 in which the authors have incorporated the proposed MATQA method into traditional or previously existing answering schemes. For example, what is the precision or hits if those traditional or previously existing schemes were used on their own without MATQA. With that comparison, it will be easier to justify the accuracy of using MATQA. Line 264 says, “…MATQA can improve the traditional…” but more explanation is required. Line 281 gives a good example of how improvement in the results can be shown.
--Line 278 is little confusing to the reader. It is little unclear whether the interference graph has a positive feedback or the boot node.
--Line 294 says “…makes up the defects of traditional top-k which cannot ….” Is there any example or explanation for this?
--Line 298 suggests that the MATQA method gives wrong results when a numerical answer is expected. However, in Table 3, it gives correct answer for the third question “When did owner Fred Wilson’s..?” What is the difference between that question and the second question in Table 4? Why is MATQA predicting correct answer for one but not for the other?

Additional comments

The authors can consider combining the Discussion and Conclusions section and remove any redundant statements. This new section can be renamed as Summary and Conclusions.

The theme of this research is relevant to PeerJ journal as it proposes a novel methodology for calculating multiple answers for temporal questions. I recommend this paper for publication, after the reviewer comments have been addressed. This manuscript can be considered for publication after the structure of sentences, typing mistakes, and reference format is followed.

---

## Round 0.2 · Minor Revisions

The reviewers are overall satisfied with the revised version of the manuscript. Nonetheless, one of the reviewers has identified some grammar and typo issues. Please address those, and in general, carry out a rigorous proof-check of your manuscript.

·

Basic reporting

Thank the authors for their efforts in considering my comments and revising the manuscript.

I have found this revised manuscript significantly improved for the previous presentation issues. All my previously mentioned points have been corrected or enhanced properly. Here, when the authors were addressing those issues, I found that the below very minor wording/grammatical issues or typos were introduced:
R1. Line 171: the the -> the
R2. Line 205: ..., here it is used to obtain the head of BERT, and ... -> ... (here it is used to obtain the head of BERT), and ...
R3. Line 216: ... is the representation of length D of node v -> ... is the representation of node v (in the form of a D-dimensional vector)

Also, after Figure 2 was adjusted (where part of the figure was enlarged), I found that:
R4. it could be slightly improved if the green arrows (which seems vague and hard to observe) could use a darker color to make them more observable.

Regarding the previous experimental issue, I have read the explanation from the authors and the revised experiment introduction which are convincing. I accept the explanation and revision about the experimental part.

Overall, I found that the quality of the manuscript has been enhanced significantly with only very minor changes to make. I would be appreciated if the editor could check with the further revisions list above (numbered from R1 to R4).

Experimental design

no comment

Validity of the findings

no comment

Additional comments

no comment

·

Basic reporting

The authors have accurately explained all of the questions by the reviewer. Corresponding relevant changes have been updated in the manuscript.

Experimental design

The authors have accurately explained all of the questions by the reviewer. Corresponding relevant changes have been updated in the manuscript.

Validity of the findings

The authors have accurately explained all of the questions by the reviewer. Corresponding relevant changes have been updated in the manuscript.

Additional comments

The authors have accurately explained all of the questions by the reviewer. Corresponding relevant changes have been updated in the manuscript.

---

## Round 0.3 · accepted · Accept

The reviewers were satisfied with the previous version, pending a few minor issues, which appear to have been addressed by the authors.